# Magnetoresistance and Thermal Transformation Arrest in Pd_2_Mn_1.4_Sn_0.6_ Heusler Alloys

**DOI:** 10.3390/ma12142308

**Published:** 2019-07-19

**Authors:** Xiao Xu, Hironari Okada, Yusuke Chieda, Naoki Aizawa, Daiki Takase, Hironori Nishihara, Takuo Sakon, Kwangsik Han, Tatsuya Ito, Yoshiya Adachi, Takumi Kihara, Ryosuke Kainuma, Takeshi Kanomata

**Affiliations:** 1Department of Materials Science, Graduate School of Engineering, Tohoku University, Sendai 980-8579, Japan; 2Faculty of Engineering, Tohoku Gakuin University, Tagajo 985-8537, Japan; 3Faculty of Science and Technology, Ryukoku University, Otsu 520-2194, Japan; 4Graduate School of Science and Engineering, Yamagata University, Yonezawa 992-8510, Japan; 5Institute for Materials Research, Tohoku University, Sendai 980-8577, Japan; 6Research Institute for Engineering and Technology, Tohoku Gakuin University, Tagajo 985-8537, Japan

**Keywords:** electric resistivity, magnetoresistance, Pd–Mn–Sn, Heusler alloy, martensitic transformation, thermal transformation arrest

## Abstract

The magnetization, electric resistivity, and magnetoresistance properties of Pd${}_{2}$Mn${}_{1.4}$Sn${}_{0.6}$ Heusler alloys were investigated. The Curie temperature of the parent phase, martensitic transformation temperatures, and magnetic field dependence of the martensitic transformation temperatures were determined. The magnetoresistance was investigated from 10 to 290 K, revealing both intrinsic and extrinsic magnetoresistance properties for this alloy. A maximum of about −3.5% of intrinsic magnetoresistance under 90 kOe and of about −30% of extrinsic magnetoresistance under 180 kOe were obtained. Moreover, the thermal transformation arrest phenomenon was confirmed in the Pd${}_{2}$Mn${}_{1.4}$Sn${}_{0.6}$ alloy, and an abnormal heating-induced martensitic transformation (HIMT) behavior was observed.

## 1. Introduction

Shape memory alloys (SMAs) are a family of functional materials showing shape memory effects (SMEs) and superelasticity (SE) via the diffusionless thermoelastic martensitic transformation. The history of SMA dates back to the discovery of SME in the Au–Cd alloy in 1951 [1]. Since the development of Ni–Ti alloys [2] in the 1960s, SMAs have not only become an important focus of research but have also become widely used in daily life, for example, in the sectors of medical care, electrical products, and space systems [3]. To date, a large number of SMA systems have been developed, such as Ti-based alloys of Ti–Pd, Ti–Pt, and Ti–Au [4]; Cu-based alloys of Cu–Zn [5] (Cu–Zn–Al [6]), Cu–Al–*X* (X=Ni [7], Be [8], and Nb [9]); and Fe-based alloys of Fe–Pt [10] and Fe–Pd [11]. Recently, some Fe-based alloys, such as Fe–Ni–Co–Ti [12], Fe–Mn–Si [13], Fe–Ni–Co–Al–Ta–B [14], and Fe–Mn–Al–Ni [15], and the Cu-based Cu–Mn–Al alloy, have been developed [16]. These systems are inexpensive and some of them show excellent ductility [15,16] and feasibility for the preparation of single crystals [17,18]; therefore, some of them have already been utilized as alternatives of the well-known Ni–Ti alloy. In Co-based Heusler ($L2_{1}$) alloys, a Co–Cr–Ga–Si system has been developed, where a unique reentrant martensitic transformation behavior and a cooling-induced SME have been realized [19]. Very recently, in a novel Mg-based system, the Mg–Sc alloy was found to show SE behavior, thus a lightweight SMA system was realized [20].

Meanwhile, the Ni–Mn-based magnetic shape memory alloys (MSMAs) with a Heusler-type structure have attracted much attention because of their potential applications in functional materials. The study of these Heusler alloys started in early pioneering works by Hames [21] and Webster et al. [22,23]; however, little notice was taken until the report of magnetic-field-induced strain by variant rearrangement in the Ni–Mn–Ga system by Ullakko et al. [24]. Since then, new MSMA systems, such as Fe–Pd [25], Fe–Pt [26], Ni–Mn–Al [27], Ni–Co–Ga [28], Ni–Co–Al [29], and Ni–Fe–Ga [30], have been successively reported. In 2006, Kainuma et al. reported a different mechanism of magnetic-field-induced strain by magnetic-field-induced transition (MFIT) behavior [31]. By the MFIT behavior, the magnetic-field-induced strain is accompanied by an output stress much greater than the case of variant rearrangement [32,33]. Furthermore, because of the occurrence of the first-order martensitic transformation during MFIT, multifunctional properties can be found, such as the metamagnetic shape memory effect [31,34,35,36], large magnetoresistance (MR) [37,38,39,40], large magnetocaloric [41,42,43,44,45], and elastocaloric [46,47,48] effects.

Among these MSMAs, Ni${}_{2}$Mn${}_{1+x}$Sn${}_{1-x}$ alloys have been widely investigated because of their robust martensitic transformation behavior and low cost. On the basis of magnetization measurements, the magnetic phase diagram of Ni${}_{2}$Mn${}_{1+x}$Sn${}_{1-x}$ (0≤x≤0.6) alloys was reported [49,50]. The results of neutron powder diffraction measurements for Ni${}_{2}$Mn${}_{1.44}$Sn${}_{0.56}$ have shown that the martensite phase has an orthorhombic structure with space group Pmma [51]. Moreover, the Ni${}_{2}$Mn${}_{1+x}$Sn${}_{1-x}$ alloys with compositions Ni${}_{2}$Mn${}_{1.44}$Sn${}_{0.56}$ [38] undergo a martensitic transformation accompanied by a large change in electric resistivity. Thus, owing to the above-mentioned MFIT behavior, an extrinsic MR of about 50% was observed [38].

Recently, a new Pd${}_{2}$Mn${}_{1+x}$Sn${}_{1-x}$ system has been developed, for which a cubic-to-orthorhombic or cubic-to-monoclinic martensitic transformation has been found [52,53]. Chieda et al. determined the magnetic phase diagram of Pd${}_{2}$Mn${}_{1+x}$Sn${}_{1-x}$ (0≤x≤0.5) [54], where the behaviors of the composition dependences of the martensitic transformation and magnetic transition temperatures were found to be similar to those of Ni${}_{2}$Mn${}_{1+x}$Sn${}_{1-x}$ [49]. However, to our knowledge, there is no information about the electrical properties or the MR of Pd${}_{2}$Mn${}_{1+x}$Sn${}_{1-x}$, except for the stoichiometric parent alloy, Pd${}_{2}$MnSn. In this study, the magnetization, electric resistivity, and magnetoresistance properties of Pd${}_{2}$Mn${}_{1.4}$Sn${}_{0.6}$ were examined to gain deeper insights into the phase transitions of Pd${}_{2}$Mn${}_{1+x}$Sn${}_{1-x}$ Heusler alloys.

## 2. Materials and Methods

Polycrystalline Pd${}_{2}$Mn${}_{1.4}$Sn${}_{0.6}$ alloy was prepared by the repeated melting of appropriate quantities of the constituent elements, namely, 99.9% Pd, 99.99% Mn, and 99.999% Sn, in an argon arc furnace. Following the repeated arc melting, the reaction product was sealed in evacuated double silica tubes. The reaction product was solution-heat-treated at 1373 K for 2 days and then annealed at 673 K for 1 day and 573 K for 3 days. The microstructure of the heat-treated sample was examined by the use of a scanning electron microscope (SEM) (JEOL Ltd., Tokyo, Japan). As shown in Figure 1, a typical single-phase microstructure was found at room temperature, with small black areas being the voids. The chemical composition of the sample was analyzed as Pd${}_{2.01\pm 0.01}$Mn${}_{1.39\pm 0.01}$Sn${}_{0.60\pm 0.01}$, where an electron probe microanalyser (EPMA) equipped with a wavelength dispersive spectroscope (WDS) (JEOL Ltd., Tokyo, Japan) was used. No noticeable macro segregation was found. The magnetization data were collected using a commercial superconducting quantum interference device (SQUID) magnetometer (Quantum Design, Inc., San Diego, CA, USA). The measurements of the electric resistance and magnetoresistance were conducted by the conventional four-probe method using the physical property measurement system (PPMS) (Quantum Design, Inc., San Diego, CA, USA), where magnetic fields up to 90 kOe were used for the measurements. The magnetoresistance at several selected temperatures were also measured using a superconducting magnet up to 180 kOe at the High Field Laboratory for Superconducting Materials, Institute for Materials Research, Tohoku University (Sendai, Japan).

## 3. Results and Discussions

### 3.1. Martensitic Transformation Behaviors

Figure 2a shows the thermomagnetization curves of the alloy. After cooling the sample to 6 K under zero magnetic field, a magnetic field of 500 Oe was applied to conduct the measurement during a heating process (zero-field-cooling (ZFC)). This was followed by field-cooling (FC) and field-heating (FH) measurements, as indicated by the arrows. A splitting behavior between ZFC and FC was observed, and the Curie temperature, $T_{C}$ was detected as 204 K; these findings are consistent with earlier results [54]. The hysteresis seen in the temperature range of about 45–135 K, which is denoted as $T_{\mathrm{FullHys}}$ in Figure 2a, originates from the martensitic transformation behavior, where the martensite phase shows a lower magnetization than the parent phase.

The temperature dependence of the electric resistivity is shown in Figure 2b. A hysteresis corresponding to the martensitic transformation was also detected. The martensitic transformation starting temperature of $T_{M_{s}}=87$ K and reverse martensitic transformation finishing temperature of $T_{A_{f}}=106$ K were determined by the extrapolation method, as shown in Figure 2b. Note that the martensitic transformation temperatures observed in Figure 2b differ slightly from those in Figure 2a. We adopted the transformation temperatures from the measurements of the electric resistivity in Figure 2b because of the possible ambiguity caused by the bending of the thermomagnetization curves near the $T_{C}$, as shown in Figure 2a. In Figure 2b, an anomaly was found near the $T_{C}$. Recently, some anomalous behaviors of the temperature dependence of the electric resistivity were found for various Heusler systems, such as a cusp behavior [55,56] and an inverse temperature dependence [57,58] near the $T_{C}$. However, the current Pd${}_{2}$Mn${}_{1.4}$Sn${}_{0.6}$ alloy shows a typical downward bending below the $T_{C}$, similar to common ferromagnetic metals such as pure Ni [59]. This is a result of the decrease in the spin-disorder scattering described by the free electron models proposed by Kasuya [60] and de Gennes and Friedel [61]. Below the $T_{C}$, a second-order polynomial,
(1)$$\rho \left(T\right)=\rho_{0}+aT+bT^{2},$$
has been widely used to fit the temperature dependence of the electric resistivity. Here, $\rho_{0}$ is the residual resistivity at 0 K and both *a* and *b* are fitting parameters. The quadratic and linear terms are presumed to be related to the magnetic and electron–phonon scatterings, respectively. However, one should note that Equation (Equation 1) is a phenomenological fit to the experimental data. Above the $T_{C}$, a linear relationship holds, and the following equation:(2)$$\rho \left(T\right)=\rho_{0}^{\prime }+cT$$
was used for the phenomenological fitting, with $\rho_{0}^{\prime }$ being the residual resistivity at 0 K and *c* being a fitting parameter. The fitting parameters, *a*, *b*, and *c*, are listed in Table 1 with earlier reported values for both the stoichiometric and non-stoichiometric Heusler alloys. It can be seen that *a* has a moderate value, whereas *b* and *c* seem to be small compared to those of other Pd-based alloys.

Figure 3a shows the temperature dependence of the magnetization under different magnetic fields up to 70 kOe. As shown by the different colors, the martensitic transformation temperatures decrease with increasing magnetic field owing to the stronger magnetization of the parent phase than that of the martensite phase. Figure 3b shows the temperature dependence of the electric resistivity under magnetic fields up to 90 kOe. Similar to Figure 3a, the martensitic transformation temperatures decrease with increasing magnetic field. In Figure 3b, the transformation temperatures are determined by the extrapolation method, similar to Figure 2b. For clarity, $T_{M_{s}}$ and $T_{A_{f}}$ are not shown individually in Figure 3b but are plotted against the the magnetic field in Figure 3c. It can be clearly seen that $T_{M_{s}}$, $T_{A_{f}}$, and the presumed equilibrium temperature $T_{0}=\left(T_{M_{s}}+T_{A_{f}}\right)/2$ exhibit a practically linear magnetic field dependence, as shown by the solid least-square linear fit lines. The slope, $dT_{0}/dH$, is about −0.74 K/10 kOe, which is a small value compared to those of Ni–Mn-based MSMAs, for which a value larger than 3 or even 10 K/10 kOe is typical [31,67,68,69]. Thus, with the consideration of the full thermal hysteresis, $T_{\mathrm{FullHys}}$, of nearly 90 K, as shown in Figure 2, to realize a full reversible MFIT at low temperatures such as 40 K, a large magnetic field of the order of 1 MOe may be required.

### 3.2. Thermal Transformation Arrest and Heating-Induced Martensitic Transformation

One may notice some anomalous behaviors at low temperatures in Figure 3a,b. Specifically, in Figure 3a, even at high magnetic fields larger than 10 kOe, the magnetization at 6 K increases obviously with increasing applied magnetic field. Similarly, in Figure 3b, the electric resistivity at 4 K decreases with increasing magnetic field. To investigate the reason for these anomalous behaviors, the following examinations were conducted.

First, as shown in Figure 4a,b, the thermomagnetization and magnetization measurements were conducted as follows:Step A.Under 0.5 kOe, a thermomagnetization measurement was conducted starting from 300 K to 6 K and back to 300 K. The magnetic field was turned off, and the temperature was set to 6 K.Step B.At 6 K, a magnetization measurement up to 50 kOe was conducted.Step C.Under 40 kOe, a magnetization measurement was conducted starting from 6 K to 300 K and back to 6 K. The magnetic field was then turned off; the temperature was maintained at 6 K.Step D.The same as Step B.Step E.Under 0.5 kOe, a thermomagnetization measurement was conducted starting from 6 K to 300 K.

It can be seen that the latter half (heating part) of Step A is identical to Step E, and Steps B and D are exactly identical. Thus, their magnetization curves should also be perfectly coincident if nothing distinct occurs during Step C. However, non-ergodic results are found for these measurements. As shown in the inset of Figure 4a, at low temperatures below about 40 K, the magnetization curves do not overlap for the heating and cooling processes. Furthermore, Steps A and E show differences below 40 K in Figure 4a, and differences can also be found in Steps B and D in Figure 4b.

This has been reported as the kinetic arrest phenomenon in Ni–Mn–In-based alloys [70,71,72,73], recently becoming known as called the thermal transformation arrest (TTA) phenomenon [69,74]. The TTA phenomenon is sometimes observed in alloy systems where the parent phase shows a larger magnetization than the martensite phase. Moreover, during the cooling process, the martensitic transformation stops below a certain temperature, resulting in an arrested parent phase at low temperatures. When the cooling process is conducted under magnetic fields, a larger amount of arrested parent phase is obtained than when it is conducted without them. At low temperatures, even when the magnetic field is turned to zero, the arrested parent phase cannot transform back to the martensite phase for kinetic reasons; thus, a heating process results in an anomalous heating-induced martensitic transformation (HIMT) [71,75].

Thus, the following explanations can be applied to understand the measurements of Steps A to E in Figure 4a,b. Step A is a thermomagnetization measurement under a negligibly weak magnetic field used as a reference. Then, Step B is conducted to record the magnetization of the nearly full martensite phase at 6 K. In Step C, during the cooling process under 40 kOe, a small amount of the parent phase is arrested; thus, a non-overlapping magnetization curve is found below about 40 K compared to that of the heating process. At this point, the magnetic field is turned off once; however, the small amount of the arrested parent phase cannot transform to the martensite phase for kinetic reasons. This can also be verified by the slightly increasing magnetization in Step D compared to Step B in Figure 4b. Finally, when the sample is heated under a negligibly weak magnetic field in Step E, the de-arrest behavior, which is also the occurrence of HIMT, can be confirmed by the decreasing magnetization with increasing temperature in the temperature range below about 50 K.

Second, the TTA phenomenon is also confirmed by the examination of the electric resistivity. By use of a strong magnetic field, the arrested parent phase is clearly confirmed. The following measurements were conducted:Step A.Under no magnetic field, the electric resistivity was measured starting from 300 K to 4 K and back to 300 K.Step B.The value of the electric resistivity was recorded at 4 K under no magnetic field.Step C.Under 90 kOe, the electric resistivity was measured starting from 300 K to 4 K.Step D.The value of the electric resistivity was recorded at 4 K under no magnetic field.Step E.Under no magnetic field, the electric resistivity was measured by heating from 4 K to 300 K.

Steps B and D were not actually performed because Step B is the value of the electric resistivity at 4 K in Step A, and Step D is the first value of Step E. The results are shown Figure 4c. The explanation is similar to Figure 4a,b. Owing to the use of a strong magnetic field in Step C, as shown in Figure 4c, a large amount of the arrested parent phase was found; thus, a very clear de-arrest behavior is observed. Summarizing, Figure 4 shows the first observation of the TTA as well as the HIMT phenomenon in Pd-based shape memory alloys.

### 3.3. Intrinsic and Extrinsic Magnetoresistance

MR (Δρ/ρ) up to 90 kOe was examined in the temperature range from 10 to 190 K; Figure 5a shows some selected results. Magnetic fields up to 90 kOe are still extremely weak in terms of resulting in an MFIT; thus, all the results in Figure 5a are intrinsic MR for the Pd${}_{2}$Mn${}_{1.4}$Sn${}_{0.6}$ alloy. In Figure 5b, the maximum value of the MR at 90 kOe is plotted against the measurement temperature. It can be seen that generally the martensite phase shows a larger value of MR than the parent phase. Within the temperature range of the parent phase, the MR shows a maximum value around the $T_{C}$, which is considered to be the cause of a large spin fluctuation around the magnetic transition temperature. At 50 K, the maximum MR was obtained to be around −3.6%. This is a moderately large value compared to the reported values of the MR of −0.65% for Co${}_{2}$Cr${}_{0.6}$Fe${}_{0.4}$Al [76], −1.7% for Ni${}_{35}$Mn${}_{50}$In${}_{15}$ [64], and −2% to −5% for Ni–Fe–Ga-based alloys [77]. However, much smaller values were obtained compared to the systems showing larger values of MR, such as around −8% for Ni${}_{2.208}$Mn${}_{0.737}$Ga [78], over −10% for Ni${}_{50}$Mn${}_{8}$Fe${}_{17}$Ga${}_{25}$ [79], and around −50% for Fe${}_{2}$VGa [80]. The reason for the large MR in the martensite phase of Pd${}_{2}$Mn${}_{1.4}$Sn${}_{0.6}$ remains unclear; however, one may notice that the high-field magnetic susceptibility, $\chi_{\mathrm{HF}}\approx 0.04$ emu·g${}^{-1}\cdot$kOe${}^{-1}$, is large, which is about twice the value of $\chi_{\mathrm{HF}}$ of the stoichiometric Pd${}_{2}$MnSn [81].

Strong magnetic fields were also used to investigate the extrinsic MR caused by the occurrence of the MFIT. Figure 6 shows the MR measured at several selected temperatures. At 150 and 4.2 K, which are in the full parent and martensite phases, respectively, only intrinsic MR was observed. At 75, 85, and 95 K, a large extrinsic MR was observed owing to the magnetic-field-induced reverse martensitic transformation from the martensite to the parent phase. At 85 K, a maximum of about −30% of extrinsic MR was observed. Note that in Figure 6, the temperatures of 75, 85, and 95 K are within the hysteresis loop of the martensitic transformation; therefore, the extrinsic MR is not reversible, as is shown by the blue arrows for 85 K. Thus, the MR will be much smaller for the second measurement if the temperatures are kept constant. A large extrinsic MR has been widely observed in NiMn-based alloys, such as in Ni–(Co)–Mn–In [37,71,82,83], Ni–(Cu)–Mn–Sn [38,84,85], and Ni–Co–Mn–Al [86]. When the change in the ρ in the martensitic transformation is very large, the value of the MR can be over −90% [71].

## 4. Conclusions

In conclusion, the electric resistivity and magnetoresistance properties were investigated for a Pd${}_{2}$Mn${}_{1.4}$Sn${}_{0.6}$ Heusler alloy showing martensitic transformation. The temperature dependence of the electric resistivity above and below the Curie temperature of the parent phase fitted well using phenomenological polynomials. The Curie temperature of the parent phase and martensitic transformation behavior were clearly observed by the electric resistivity measurements, and the martensitic transformation temperatures decreased with the increasing magnetic field. The thermal transformation arrest phenomenon was observed in the current alloy, and the unique heating-induced martensitic transformation behavior was also confirmed. The magnetoresistance was also investigated in a wide temperature range. A maximum intrinsic magnetoresistance of about −3.6% was found at 50 K under a magnetic field of 90 kOe, whereas a maximum extrinsic magnetoresistance of about −30% was found at 85 K owing to the magnetic-field-induced reverse martensitic transformation.

## Figures and Tables

**Figure 1 materials-12-02308-f001:**
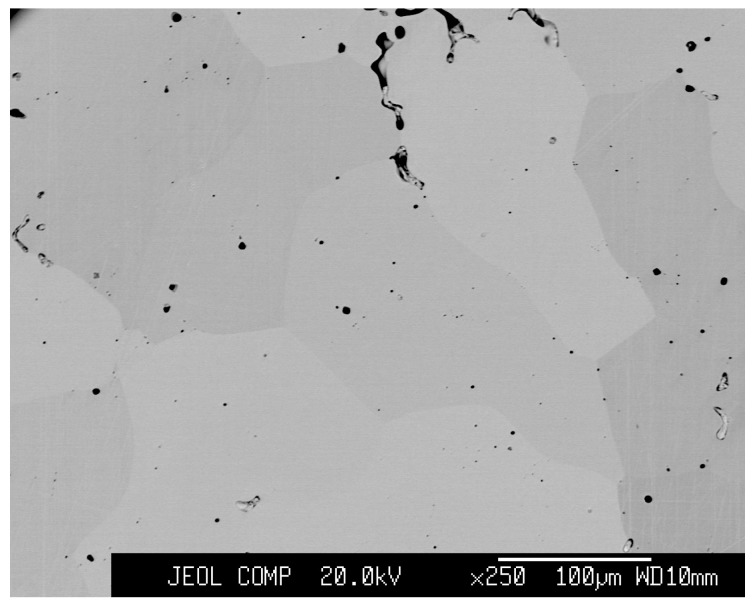
SEM micrograph taken using back-scattered electrons for Pd${}_{2}$Mn${}_{1.4}$Sn${}_{0.6}$ at room temperature.

**Figure 2 materials-12-02308-f002:**
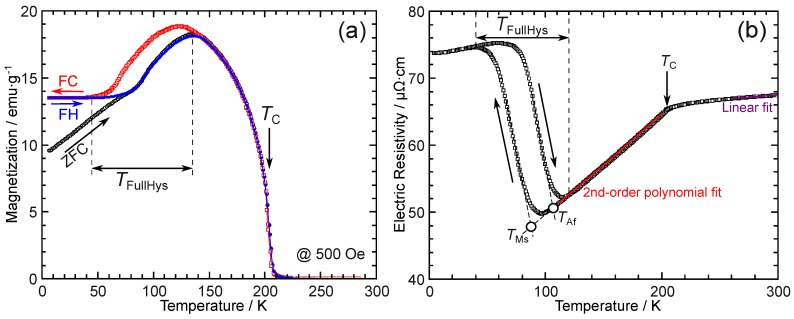
Results of (**a**) thermomagnetization measurements under 500 Oe and (**b**) temperature dependence of the electric resistivity of Pd${}_{2}$Mn${}_{1.4}$Sn${}_{0.6}$. In the figures, the Curie temperature is denoted as $T_{C}$. In (**a**), the zero-field-cooling (ZFC), field-cooling (FC), and field-heating (FH) measurements are presented. In (**b**), $T_{M_{s}}$ and $T_{A_{f}}$ indicate the martensitic transformation starting and reverse transformation finishing temperatures, respectively, with $T_{\mathrm{FullHys}}$ indicating the full thermal hysteresis of the thermoelastic martensitic transformation. For the parent phase, linear and second-order polynomial fits are conducted for the temperature ranges above and below the $T_{C}$, respectively. Refer to the text for details.

**Figure 3 materials-12-02308-f003:**
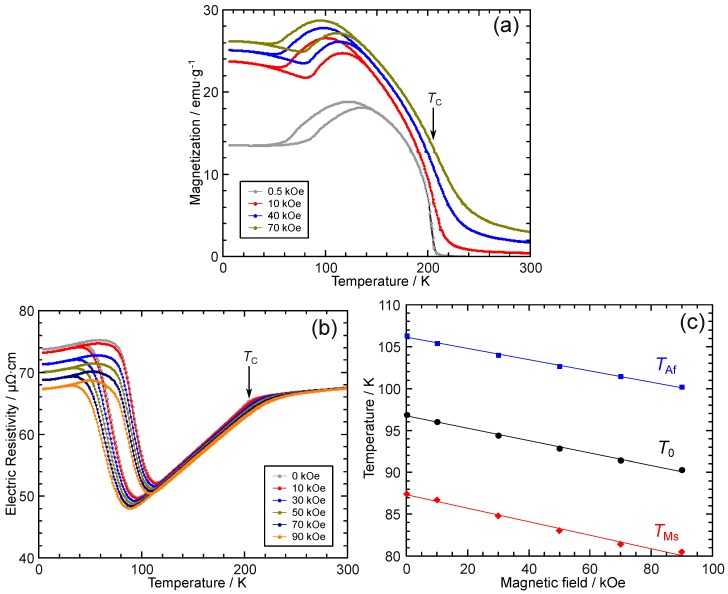
(**a**) Temperature dependence of the magnetization under magnetic fields of 0.5, 10, 40, and 70 kOe. (**b**) Temperature dependence of the electric resistivity under magnetic fields of 10, 30, 50, 70, and 90 kOe. (**c**) The martensitic transformation starting temperature, $T_{M_{s}}$, reverse martensitic transformation finishing temperature, $T_{A_{f}}$, and equilibrium temperature supposed as $T_{0}=\left(T_{M_{s}}+T_{A_{f}}\right)/2$ determined from (**b**) are plotted against the magnetic field. Solid lines are least-square linear fits.

**Figure 4 materials-12-02308-f004:**
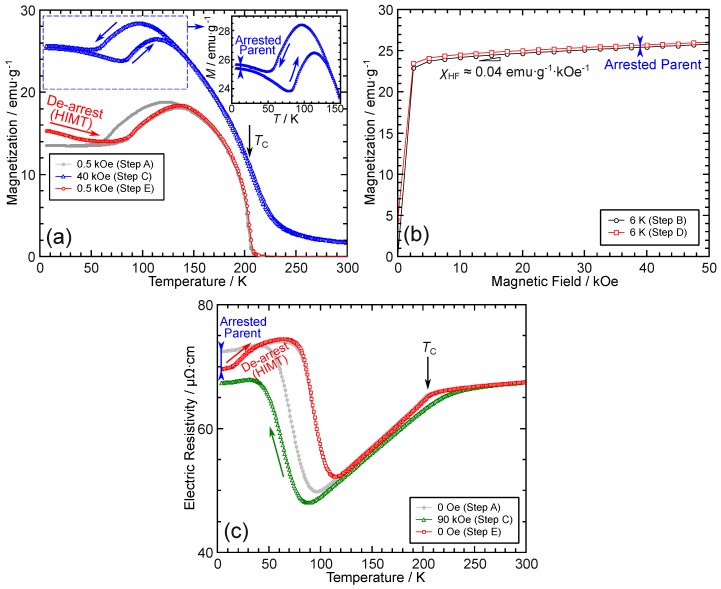
(**a**) Thermomagnetization, (**b**) magnetization, and (**b**) electric resistivity measurements intended for the observation of the thermal transformation arrest (TTA) phenomenon. The de-arrest behavior, where the heating-induced martensitic transformation (HIMT) occurs, was observed in both (**a**) and (**c**). Refer to the text for the details and the definitions of Steps A to E. In (**b**), $\chi_{\mathrm{HF}}$ denotes the high-field magnetic susceptibility of the martensite phase.

**Figure 5 materials-12-02308-f005:**
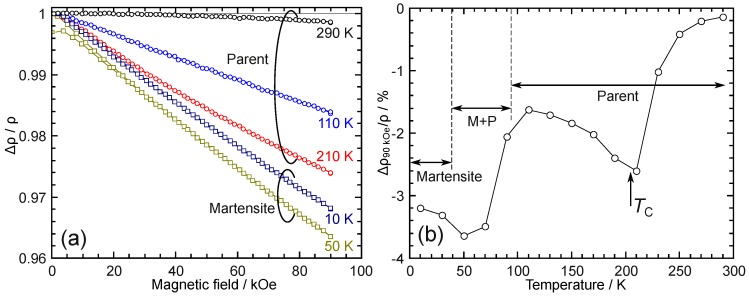
(**a**) Intrinsic magnetoresistance measured up to 90 kOe at 290, 210, 110, 50, and 10 K. (**b**) Intrinsic magnetoresistance under 90 kOe plotted against the measurement temperature. A peak is found near the Curie temperature of the parent phase, $T_{C}$. The martensite phase generally shows a higher magnetoresistance than the parent phase.

**Figure 6 materials-12-02308-f006:**
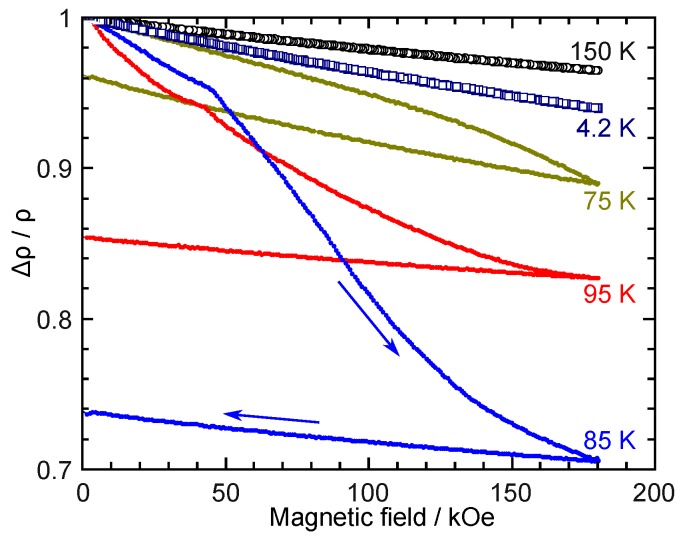
Change in the electric resistivity against magnetic fields up to 180 kOe. At 4.2 and 150 K, only intrinsic magnetoresistance is seen. However, a magnetic-field-induced transition is found at other temperatures. A maximum of about 30% of extrinsic magnetoresistance is observed at 85 K.

**Table 1 materials-12-02308-t001:** Curie temperature $T_{C}$, fitting parameters *a*, *b*, and the fitting range obtained by use of Equation (Equation 1), and *c* by use of Equation (Equation 2) for the Pd${}_{2}$Mn${}_{1.4}$Sn${}_{0.6}$ alloy. Earlier reports for other Pd-, Ni-, and Co-based stoichiometric and non-stoichiometric Heusler alloys are listed for comparison.

Alloy	$T_{C}$ (K)	Second-Order Polynomial Fit				Linear Fit	Reference
		$a\times 10^{-2}$	$b\times 10^{-4}$	Range (K)		$c\times 10^{-2}$
Pd${}_{2}$Mn${}_{1.4}$Sn${}_{0.6}$	204	12.6	0.79	110–200		1.35	This work
Pd${}_{2}$MnIn	-	-	-	-		2.0	[62]
Pd${}_{2}$MnSn	189	9.2	6.1	105–180		7.1	[62]
Pd${}_{2}$MnSb	247	11.4	5.3	105–240		10.8	[62]
Ni${}_{2}$MnIn	323	2.71	4.23	150–300		3.3	[62]
	320–321	1.57–1.68	4.29–4.43	140–300		-	[63]
Ni${}_{2}$MnSn	344	4.46	3.94	150–300		-	[62]
Ni${}_{2}$MnSb	334	10.4	2.73	150–300		-	[62]
Ni${}_{50}$Mn${}_{36}$In${}_{14}$	325	33.4	3.19	220–280		-	[38]
Ni${}_{35}$Mn${}_{50}$In${}_{15}$	333	25.4	1.34	100–300		2.68	[64]
Ni${}_{2}$MnGa${}_{1-x}$Fe${}_{x}$	377–431	3.34–9.18	0.26–6.37	100–390		-	[65]
Co${}_{2}$TiAl	126	70	11	-		42.7	[66]
Co${}_{2}$Ti${}_{1.25}$Al${}_{0.75}$	155	70	10	-		39.4	[66]
Co${}_{2}$ZrAl	194	60	17.6	-		42.2	[66]
Co${}_{2}$Zr${}_{1.15}$Al${}_{0.85}$	196	140	39.5	-		64.7	[66]
Co${}_{2}$Zr${}_{0.85}$Al${}_{1.15}$	188	80	0.5	-		44.8	[66]

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
