# Peer review of "Magnetoresistance and Thermal Transformation Arrest in Pd2Mn1.4Sn0.6 Heusler Alloys"

_materials, 2019, doi:10.3390/ma12142308_

Reviewer 1 Report

The manuscript is well written and structured. The discussion is sound.

Author Response

Thank you very much for your positive comments.

Reviewer 2 Report

Dear Editor,

in my opinion the article is of very good quality and can be published as it is presented.

Yours faithfuly,

Rafał Wróblewski

Author Response

(The authors gave the same response as above.)

Reviewer 3 Report

congratulations for the high quality work. however, there might be some minor modifications prior to publishing the paper that can strengthen the work:

the introduction is not complete and it must contain different aspects that are involved in your work. as an example, you can simply introduce shape memories in s couple of sentences and add more recent publications to upgrade the introduction. in its current format, the introduction is not acceptable at all. the following articles are strongly recommended:

https://link.springer.com/article/10.1007/s10853-019-03375-1

https://www.sciencedirect.com/science/article/pii/S0924013619301165

https://www.sciencedirect.com/science/article/abs/pii/S135964541830911X

https://www.sciencedirect.com/science/article/abs/pii/S1359645419300114

since the shape memory is polycrystalline, it would be useful to have at least one image of the surface of the materials that you prepared by arc melting. in a paper that is on a materials science topic, at least one image of the surface is mandatory. please add one.

did you do any analysis on the chemical composition of the final product you prepared? the melted materials are not necessarily the same composition as their ingredients before melting. you have to do some characterization tests and make sure that there are no other phases or precipitates that alter the properties of the shape memory you are studying. maybe and XRD test will do this need.

Author Response

Thank you very much for your constructive comments and suggestions. Attached please see the response letter where your requests are answered one by one.

Round  2

Reviewer 3 Report

Thank you for revising the manuscript.

One of my major comments was the old references which are not addressed well. However, I agree with the other reviewers that the current format is good.